# Polymorphisms of DNA Repair Genes in Thyroid Cancer

**DOI:** 10.3390/ijms25115995

**Published:** 2024-05-30

**Authors:** Adrianna Gielecińska, Mateusz Kciuk, Damian Kołat, Weronika Kruczkowska, Renata Kontek

**Affiliations:** 1Department of Molecular Biotechnology and Genetics, Faculty of Biology and Environmental Protection, University of Lodz, Banacha Street 12/16, 90-237 Lodz, Poland; adrianna.gielecinska@edu.uni.lodz.pl (A.G.); renata.kontek@biol.uni.lodz.pl (R.K.); 2Doctoral School of Exact and Natural Sciences, University of Lodz, Banacha Street 12/16, 90-237 Lodz, Poland; 3Department of Functional Genomics, Medical University of Lodz, 90-752 Lodz, Poland; damian.kolat@umed.lodz.pl; 4Department of Biomedicine and Experimental Surgery, Medical University of Lodz, 90-136 Lodz, Poland; 5Faculty of Biomedical Sciences, Medical University of Lodz, Zeligowskiego 7/9, 90-752 Lodz, Poland; weronika.kruczkowska@stud.umed.lodz.pl

**Keywords:** DNA damage response, DNA repair genes, follicular thyroid cancer, papillary thyroid cancer, polymorphism

## Abstract

The incidence of thyroid cancer, one of the most common forms of endocrine cancer, is increasing rapidly worldwide in developed and developing countries. Various risk factors can increase susceptibility to thyroid cancer, but particular emphasis is put on the role of DNA repair genes, which have a significant impact on genome stability. Polymorphisms of these genes can increase the risk of developing thyroid cancer by affecting their function. In this article, we present a concise review on the most common polymorphisms of selected DNA repair genes that may influence the risk of thyroid cancer. We point out significant differences in the frequency of these polymorphisms between various populations and their potential relationship with susceptibility to the disease. A more complete understanding of these differences may lead to the development of effective prevention strategies and targeted therapies for thyroid cancer. Simultaneously, there is a need for further research on the role of polymorphisms of previously uninvestigated DNA repair genes in the context of thyroid cancer, which may contribute to filling the knowledge gaps on this subject.

## 1. Introduction

Genetic variation is a crucial factor in the evolution and survival of species, enabling adaptation to changing environmental conditions and providing resistance to diseases and harmful agents. Among the key aspects of genetic variability, gene polymorphisms and their impact on the risk of the development of diseases have been the subject of intensive research for many years. Gene polymorphism is a naturally occurring phenomenon within genes characterized by variability in DNA sequences between individuals in a population of a given species. This means that at a specific gene position, there may be more than one possible nucleotide sequence. Gene polymorphisms are usually inherited and constitute a natural part of genetic variability in the population. In contrast to decidedly rarer mutations, genetic polymorphism is established when the occurrence of two or more alleles at a given locus in the population reaches a frequency above 1%.

There are several types of gene polymorphism, with single nucleotide polymorphisms (SNPs) being the most commonly studied. SNPs involve a change in a single nucleotide of the DNA sequence and include substitutions (replacement of one nucleotide with another), insertions (addition of one nucleotide), or deletions (removal of one nucleotide) [1]. Among other types are microsatellite polymorphisms—Short Tandem Repeat (STR)—and minisatellite polymorphisms—Variable Number Tandem Repeat (VNTR)—which result from differences in the number of shorter or longer DNA sequence repetitions [2]. Polymorphism may also involve repeated sequences of hundreds or even thousands of nucleotides. Changes in the number of repetitions can affect gene expression and be associated with various disease states [3].

Variability in the coding sequence of genes can influence the structure of proteins and their functions [4]. Moreover, polymorphism in non-coding regions such as introns or regulatory regions may affect transcription, translation, and gene regulation processes. Understanding various types of genetic polymorphisms is crucial in genetics, evolutionary biology, personalized medicine, and research related to the inheritance of diseases. STR and VNTR are widely used in forensic genetics for DNA profiling and genetic diagnostics to construct genetic maps or determine gene carrier status [3,5,6]. However, the variability of the human genome is primarily attributed to SNPs, which are essential in studies related to the inheritance of traits, the predisposition to diseases, and personalized medicine. Many SNP polymorphisms occur in genes related to DNA repair processes, which are crucial for maintaining genome integrity [7,8]. These genes encode proteins that are part of DNA repair systems, i.e., the DNA damage response (DDR) pathway, base excision repair (BER), nucleotide excision repair (NER), DNA mismatch repair (MMR), single-strand break repair (SSBR), and double-strand DNA break repair (DSBR) by homologous recombination (HR) or non-homologous end joining (NHEJ) [9]. Polymorphisms in genes encoding repair proteins can affect their structure, function, or ability to effectively repair DNA. As a result, individuals carrying specific allele variants may have different DNA repair efficiencies compared to other individuals in the population. It is believed that abnormalities in DNA repair processes related to gene polymorphisms may constitute an important risk factor for the development of cancer. These abnormalities can lead to the accumulation of genetic mutations, which in turn promote uncontrolled cell growth and tumor development [7].

Previous research has suggested that certain polymorphisms in genes related to DNA repair processes, cell cycle regulation, or apoptosis may influence the susceptibility of individuals to thyroid cancer. Due to the limited number of up-to-date review papers that address the relationship between DNA repair gene polymorphisms and thyroid cancer susceptibility in different populations, we decided to focus on this topic. This article collects and comprehensively discusses information on the polymorphisms of selected DNA repair genes in thyroid cancer based on the available literature in the PubMed, Elsevier, Springer, and Google databases. The present manuscript used original research and review papers related to the topic described, published mainly in the last 15 years. Furthermore, we endeavored to analyze and contrast these genetic variations among other populations for which we were able to gather data.

## 2. Thyroid Cancer, DNA Damage, and Repair

Thyroid cancer is the most common malignant tumor in people aged 16 to 33 and the ninth most common cancer in the world [10]. The disease affects women more often than men. The neoplasm develops from the cells of the thyroid gland, a butterfly-shaped gland located at the base of the neck, right below the Adam’s apple. The thyroid gland produces hormones that control metabolism and are essential for numerous body functions, such as regulating the cardiovascular system by affecting heart rate, rhythm, and myocardial contraction; controlling growth, development, and maintenance of bones in adults; and influencing the production of other hormones by the endocrine glands [11,12,13]. Thyroid cancer refers to various malignant tumors of the thyroid gland, with clinical presentation, therapy, and prognosis being determined by particular tumor subtypes. For example, papillary thyroid cancer (PTC) is the most common type that constitutes approximately 80% of cases [14]. Generally, PTC presents a favorable prognosis and tends to exhibit slow growth. It is typically confined to one lobe of the thyroid but may spread to nearby lymph nodes [14,15]. In contrast, follicular thyroid cancer (FTC) represents 10–15% of thyroid cancer cases. Predominantly, it affects older adults and has the potential to spread to blood vessels or distant organs. The thyroid cancer treatment involves surgery (thyroidectomy as partial or total removal of the gland or lymph node removal if cancer has spread), radioactive iodine therapy (used to eliminate any remaining thyroid tissue or cancer cells after surgery), hormone replacement therapy (necessary after thyroidectomy to replace the hormones normally produced by the thyroid and external beam radiation), and chemotherapy (employed in advanced or aggressive cases) [16]. The prognosis for thyroid cancer is generally favorable, especially for well-differentiated types like papillary and follicular. Early detection and appropriate treatment significantly improve outcomes [17].

DNA damage is a significant factor that initiates tumor formation. In response to constant exposure of cells to DNA damaging factors, cells evolve the DDR pathways that involve many distinct repair mechanisms that help preserve genomic stability and integrity. DDR machinery abnormalities are linked to various cancer types, including thyroid cancer. Analyzed RNA sequencing data from 95 human thyroid tumor specimens showed that FTC and PTC have a higher activation level of DNA repair pathways compared to follicular adenomas [18]. The findings indicate that papillary malignancies exhibit higher levels of activated DNA repair pathways compared to follicular neoplasms, except for the G2M transition checkpoint pathway, which displayed a contrasting pattern. Furthermore, the study indicates that DNA repair capacity is higher in malignant than in benign tumors. Revealing these distinctions can enhance comprehension of the molecular processes involved in thyroid tumor development and aid in the advancement of targeted therapy strategies for various types and subtypes of thyroid cancers. In contrast, no distinct DNA repair mechanism characteristics were identified in tumor samples that were resistant to radioiodine (compared to those that were sensitive). The biological distinctions across thyroid tumor types may be more substantial than the variances between radioiodine-resistant and radioiodine-sensitive cancers [18]. Additionally, the study by Qin et al. has found that radiation exposure is linked to mutations in genes responsible for repairing double-strand breaks (DSBs), thereby reducing the ability to repair DSBs and resulting in papillary thyroid microcarcinoma (PTMC) [19]. Similarly, dysfunctions in DNA repair were demonstrated to be significant in the clonal evolution of thyroid carcinoma [20].

The DDR pathway is a critical cellular mechanism that monitors, detects, and repairs DNA damage. DNA damage can occur due to various factors such as exposure to radiation, chemicals, UV light, and errors during DNA replication. Unrepaired or improperly repaired DNA damage can lead to mutations and genomic instability and potentially contribute to the development of diseases like cancer. The pathway involves a series of molecular events orchestrated by a network of proteins including DNA repair protein (XRCC)-1, XRCC2, XRCC3, XRCC4, XRCC5, XRCC7, cellular tumor antigen p53 (P53), DNA repair protein RAD51 homolog 1 (RAD51), DNA repair protein RAD52 homolog (RAD52), breast cancer type 1 susceptibility protein (BRCA1), breast cancer type 2 susceptibility protein 2 (BRCA2), apurinic/apyrimidinic endonuclease (APE1), poly(ADP-ribose) polymerase-1 (PARP-1), 8-oxoguanine-DNA glycosylase 1 (OGG1), adenine DNA glycosylase (MUTYH), ataxia telangiectasia mutated (ATM), and checkpoint kinase 2 (CHK2). These proteins play critical roles in various stages of DNA damage recognition, signaling, and repair. ATM senses DSBs and initiates signaling cascades activating downstream transductors such as CHK2 and effectors, including TP53, to halt the cell cycle and promote repair [21,22]. In contrast, RAD52 facilitates the assembly of RAD51 nucleoprotein filaments, promoting homology search and strand invasion during HR repair in cooperation with BRCA2. XRCC5 and XRCC7 are essential components of the NHEJ pathway, repairing DSBs by directly ligating broken DNA ends with DNA ligase IV, which finalizes the repair process [23,24]. PARP-1 detects single-strand breaks (SSBs) and facilitates repair by recruiting repair factors. It plays a role in the repair of DNA lesions and interacts with various DDR pathway components. XRCC1 is involved in the repair of SSBs and base lesions via BER, where it serves as a scaffold for other protein system components. In contrast, OGG1, MUTYH, and APE1 are specific enzymatic components of BER/SSBR pathways and are involved in the repair of oxidative DNA damage by recognizing and removing modified bases and repairing apurinic/apyrimidinic sites (AP sites) generated during their excision (Figure 1) [21,22,25].

## 3. Polymorphism of Genes Encoding DNA Repair Proteins in Thyroid Cancer

### 3.1. X-ray Repair Cross-Complementing Group 1 (XRCC1)

XRCC1 is a 633 amino acid multi-domain protein that plays a key role in DNA repair processes, including BER and SSBR. It is necessary for survival, and its depletion leads to the death of mouse embryos [26]. The XRCC1 protein is encoded by the *XRCC1* gene located on chromosome 19q13.2. It is involved in SSBR, where it interacts with enzymatic components such as DNA kinases, phosphatases, polymerases, deadenylases, and ligases. Many SNPs of this protein are known, with the main ones being rs1799782 (Arg194Trp), rs25489 (Arg280His), and rs25487 (Arg399Gln), which are the subjects of intensive research related to the risk of cancer in various populations. A review of several different studies conducted in countries such as Iran, Pakistan, Saudi Arabia, China, Korea, the Czech Republic, and Portugal provides valuable information on the relationship between *XRCC1* polymorphisms and predisposition to thyroid cancer [27,28,29,30,31,32,33,34,35,36,37].

In the case of studies conducted on the Pakistani population, the heterozygous (GA) rs25489 genotype did not show a significant association with thyroid cancer, while the homozygous mutant (AA) of the same SNP significantly reduced the risk of thyroid diseases in patients compared to the control group (odds ratios (OR) = 0.17; 95% confidence interval (CI) = 0.10–0.31; *p* = 0.0001) [27]. In Iran, rs25489 analysis showed a 1.42-fold increase in the risk of differentiated thyroid cancer (DTC) demonstrated for the dominant feature GG + GA compared to AA (OR = 1.42; 95% CI = 0.76–2.68; *p* = 0.275), but the result did not reach statistical significance [28]. However, in the Spanish population, among all tested *XRCC1* polymorphisms, a positive correlation was obtained for rs25489 in the dominant trait (OR = 1.58; 95% CI = 1.05–2.46; *p* = 0.027) and nearly in the codominant model (OR = 1.61; 95% CI = 1.05–2.46; *p* = 0.087) [36]. Analyses of Chinese populations did not reveal statistically significant differences between genotypes for the tested polymorphism [30,31,32]. Studies on the rs25487 polymorphism showed a 50% reduction in the risk of DTC for the homozygous AA variant in the studied non-Hispanic white population (OR = 0.5; 95% CI = 0.3–0.8; *p* = 0.007) and in the Chinese population (OR = 0.49; 95% CI = 0.28–0.84, *p* = 0.01) [31,37]. However, Wang et al., Chiang et al., and Ryu et al., researching the Chinese and Korean populations, did not obtain statistically significant results on the role of the rs25487 polymorphism and susceptibility to thyroid cancer [30,32,33]. Similarly, research by Santos et al. on the Portuguese population did not show a significant impact of the rs25487 polymorphism on susceptibility to thyroid cancer [35]. Analysis of the rs1799782 polymorphism in the Pakistani population showed that the homozygous mutant (TT) significantly reduced the risk (OR = 0.71; 95% CI = 0.50–1.01; *p* = 0.05) of the disease in patients compared to the control group [27]. Fard-Esfahani et al. reported that in an Iranian population, rs1799782 increased the risk of developing DTC by 3.6-fold in the case of the homozygous TT genotype compared to wild-type AA (OR = 3.66; 95% CI = 0.38–35.60; *p* = 0.226); however, this result was not statistically significant [28]. In all analyzed studies on the Chinese population, it was found that the TT genotype of the rs1799782 polymorphism was associated with an approximately two-fold increased risk of thyroid cancer [30,31,32], while in the non-Hispanic white population, it showed a more than 10-fold increased risk (OR = 10.4; 95% CI = 1.0–10.5; *p* = 0.048) [37]. In the Korean population, the heterozygous AT genotype showed a two-fold reduction in the risk of this disease (OR = 0.550; 95% CI = 0.308–0.983; *p* = 0.044) [33]. Studies by Siraj et al. on the population of Saudi Arabia did not show a significant relationship between the tested polymorphisms and thyroid cancer [29]. In the Czech cohort, another *XRCC1* rs3213245 polymorphism (T-77C) influenced the genetic susceptibility to the development of papillary thyroid cancer in men [34]. However, the authors did not observe a significant association of the *XRCC1* rs1799782, rs25489, and rs25487 polymorphisms with thyroid cancer. Multiple variants of *XRCC1* can interact with variants of other DNA repair genes, such as *XRCC3*, significantly increasing susceptibility to DTC. The results of the above study suggest a differential relationship between *XRCC1* polymorphisms and the risk of thyroid cancer depending on the study population, which highlights the complexity of the influence of genetics on the predisposition to this type of cancer. The genotypic frequencies of selected SNPs of the *XRCC1* gene in various populations and their associations with the risk of thyroid cancer are presented in Table 1.

The results of several meta-analyses yielded conflicting results, suggesting the complexity of the association between *XRCC1* polymorphisms and thyroid cancer risk depending on ethnicity. Mandegari et al. conducted a meta-analysis on polymorphisms in the *XRCC1* gene—rs25487, rs25489, and rs1799782—and did not demonstrate a significant association with the risk of thyroid cancer [38]. However, subgroup analysis suggested an association of rs25487 with the risk of thyroid cancer in Caucasians but not in the Asian population. Liu et al. showed that the C allele of rs1799782 was associated with a significant reduction in the risk of thyroid cancer in Chinese people, but no significant associations were found in Caucasians [39]. However, Zhao et al. reported that the rs1799782 polymorphism was associated with the risk of thyroid cancer among Caucasians but not Asians [40]. Wang et al. found that rs25489 and rs1799782 were associated with increased risk and rs25487 with decreased risk in a Caucasian population [41]. In the most recent research from 2023, Yang et al. found a lower percentage of rs25489 GA and AA polymorphisms in thyroid cancer patients compared to the control group, but only GA showed statistical significance [42]. The discrepancies may result from differences in the studied populations, the research methodology used, and the assessment of genotypes. More well-designed trials are needed, taking into account ethnic and environmental differences, to better understand the impact of *XRCC1* polymorphisms on the development of thyroid cancer.

### 3.2. X-ray Repair Cross-Complementing Group 2 (XRCC2)

XRCC2 is a 280 amino acid protein involved in the double-stranded DNA HR repair pathway. Moreover, it is involved in the maintenance of genome integrity and control of genome rearrangement processes. It is encoded by the *XRCC2* gene, a paralog of *RAD51*, located on human chromosome 7q36.1. In exon 3, an rs3218536 polymorphism (Arg188His) was identified as a potential cancer susceptibility locus. The results of these associations remain controversial. Several different studies have been conducted to examine the association between *XRCC2* polymorphism and thyroid cancer in Iranian, Portuguese (Caucasian), Spanish (Caucasian), and Chinese populations [31,36,43,44].

A possible biological link between XRCC2 and cancer has previously been suggested, considering that a non-conservative substitution or deletion of amino acid 188 can significantly affect cellular sensitivity to DNA damage [36]. However, studies conducted in Spain and Portugal on Caucasians did not show a significant association of the rs3218536 polymorphism with the risk of thyroid cancer [36,44]. Similarly, the *XRCC2* polymorphism analysis carried out in the Chinese population did not show a significant difference in allele and genotype frequencies between the control group and the study group [31]. The results of the analysis of *XRCC2* polymorphisms also did not show a significant association with the risk of thyroid cancer in studies conducted in the Iranian population [43]. However, a polygenic risk analysis, taking into account genotype combinations of different DNA repair genes, including *RAD52* (CC)/*XRCC2* (GG)/*XRCC3* (CT + TT) (OR = 2.204; 95% CI = 1.08–4.51; *p* = 0.029), have indicated positive association where the combined variant *RAD52*/*XRCC2*/*XRCC3* showed a significant five-fold increase in the risk of thyroid cancer compared to the wild type (OR = 5.04; 95% CI = 1.45–17.58; *p* = 0.007) [43]. However, the overall and combined meta-analyses showed that the *XRCC2* rs3218536 polymorphism was not significantly associated with the risk of thyroid cancer [38,45]. The genotypic frequencies of selected SNPs of the *XRCC2* gene in various populations and their associations with the risk of thyroid cancer are presented in Table 2.

### 3.3. X-ray Repair Cross-Complementing Group 3 (XRCC3)

The *XRCC3* gene encodes a 346 amino acid protein that plays an important role in the repair of DSBs through the HR repair pathway, where it participates in the assembly and stabilization of the RAD51 protein. A specific segment of the XRCC3 protein, spanning amino acid residues 63 to 346, was identified as the region that interacts with the RAD51C protein. The amino acid residues Tyr139 and Phe249 are particularly important for the XRCC3-RAD51C interaction [46]. Polymorphisms in these regions may hinder the interactions of these proteins, contributing to the weakening of repair systems functionality and increasing the risk of cancer. Moreover, XRCC3 is involved in the repair of damage related to fragmentation, translocation, and deletion in chromosomes. Many studies have been conducted examining the effect of various *XRCC3* polymorphisms on susceptibility to cancer [29,31,36,43,44,47].

In the various populations, polymorphisms in the *XRCC3* gene showed conflicting effects on the risk of thyroid cancer. In a study conducted in Spain in the Caucasian population, analysis of the rs1799796 polymorphism (IVS7-14) did not show strong statistical associations with the risk of thyroid cancer, although a weak protective effect was suggested in combination with *XRCC2* rs3218536 (Arg188His) [36]. However, these results were not statistically robust due to the small number of individuals with a given rs3218536 A>G genotype combination. In the Chinese population, the rs1799796 polymorphism also showed no statistically significant differences between the control and random groups; neither did the rs1799794 and rs56377012 polymorphisms [31]. However, the same study found that the combination of the *XRCC1* rs25487 and *XRCC3* rs861539 genotypes contributed to an increased risk of thyroid cancer. This was a 3.08-fold increase in the risk of DTC for the combination of GG homozygous in rs25487 and TT homozygous in rs861539 (OR = 3.08; 95% CI = 1.32–7.180; *p* = 0.008); a 3.66-fold increased risk of DTC for CT heterozygous in rs1799782 and TT homozygous in rs861539 (OR = 3.66; 95% CI = 1.476–9.091; *p* = 0.005); a 3.13-fold increase in DTC risk for CC homozygous in rs1799782 and TT homozygous in rs861539 (OR = 3.13; 95% CI = 1.343–7.277; *p* = 0.008); and a 2.54-fold increase in the risk of DTC for the mutant homozygous in rs1799782 and heterozygous rs861539 heterozygote (OR = 2.54; 95% CI = 1.057–6.106; *p* = 0.039) [31]. In a Portuguese study in the Caucasian population, the rs861539 polymorphism was associated with a higher risk of thyroid cancer, especially in the case of homozygous genotype (MM) (OR = 2.0; 95% CI = 1.1–3.6; *p* = 0.026) [44]. Moreover, it was discovered that the interaction of allele variants of the *XRCC3* rs861539 and *RAD51* rs1801321 genes may significantly influence the risk of developing thyroid cancer, depending on the histological subtype of the tumor. The co-occurrence of three variant alleles of *XRCC3* rs861539 and *RAD51* rs1801321 was associated with a 2.9-fold higher incidence of primary cancer (adjusted OR = 2.9; 95% CI = 1.1–7.7; *p* = 0.036), and in the case of the presence of four variant alleles of any gene, up to eight times (adjusted OR = 8.0; 95% CI = 1.8–35.0; *p* = 0.006) [44]. However, in a non-Hispanic white population, Sturgis et al. observed an increased risk of DTC in heterozygotes (adjusted OR = 2.1; 95% CI = 1.2–3.5; *p* = 0.006) [47]. In the Iranian population, the rs861539 polymorphism in *XRCC3* showed a significant association with the risk of thyroid cancer [43]. The results showed that the CT + TT genotype, compared to the CC genotype, was associated with a 1.58-fold increased risk of developing thyroid cancer (OR = 1.58; 95% CI = 1.03–2.42; *p* = 0.035). An increased risk of DTC occurred in the case of a combination of *RAD52*, *XRCC2*, and *XRCC3* variants, as we wrote in Section 3.2. In contrast, a study conducted in Saudi Arabia found no significant association of rs861539 with the risk of developing thyroid cancer, except for the CT genotype, which suggested a non-significant reduced risk [29]. Notable differences in the impact of *XRCC3* polymorphisms on thyroid cancer risk between the studied populations suggest that genetic and environmental influences play a key role in this context. The genotypic frequencies of selected SNPs of the *XRCC3* gene in various populations and their associations with the risk of thyroid cancer are presented in Table 3.

The results of meta-analyses regarding polymorphisms of the *XRCC3* gene in thyroid cancer are mixed. According to Mandegari et al., rs1799796 and rs861539 in *XRCC3* did not show a significant relationship with the risk of thyroid cancer [38]. The ethnic subgroup analyses did not confirm a significant association between rs861539 and the risk of thyroid cancer in Asians and Caucasians. In the meta-analysis by Lu et al., the results were more complex [48]. The rs861539 polymorphism showed a positive association with the risk of thyroid cancer throughout the analysis, especially in the recessive model and in the homozygous comparison. However, analysis of ethnic subgroups did not confirm a significant relationship in individual groups. In a meta-analysis, Yu et al. found no association between *XRCC3* rs861539 and the risk of thyroid cancer in the general population [49]. The subgroup analysis in Caucasians showed a significant association, while that of Asians did not show an association with the risk of thyroid cancer. These results suggest that the effect of *XRCC3* polymorphisms on the risk of thyroid cancer may be related to ethnic differences. More studies are needed, taking into account genetic and environmental differences, to better understand the complexity of the association between *XRCC3* polymorphisms and thyroid cancer risk in different populations.

### 3.4. X-ray Repair Cross-Complementing Group 4 (XRCC4)

The *XRCC4* gene is located on human chromosome 5 and encodes the 336 amino acid XRCC4 protein, which is involved in DNA repair mechanisms, especially in the NHEJ. Additionally, the XRCC4 protein is a key component of the XRCC4/LIG4 protein complex, which allows the ends of broken DNA strands to be joined, ensuring genome integrity. *XRCC4* polymorphisms have been identified and subjected to epidemiological studies to assess their potential association with predisposition to various diseases, including cancer [50,51,52].

In DNA repair-related studies on thyroid cancer, *XRCC4* polymorphisms were analyzed in Portuguese, Chinese, and Arab populations. No study showed a significant association between the tested polymorphisms and the risk of thyroid cancer. Gomes et al. examined the rs1805377 (Asn298Ser) and rs28360135 (Thr134Ile) polymorphisms of *XRCC4* and their interactions with *LIG4*; however, no significant gene-gene interaction was demonstrated, and the analysis of the polymorphisms did not indicate a relationship with the risk of cancer in the study population [53]. In the Arab population, the analysis of *XRCC4* rs1805377 also did not show a significant statistical correlation with the risk of thyroid cancer compared to the control groups [29]. Similarly, in a study on the Chinese population, analysis on rs2035990 *XRCC4* and interactions with other genes related to DNA repair did not show statistical significance in thyroid cancer risk [54]. The genotypic frequencies of selected *XRCC4* SNPs in various populations and their associations with the risk of thyroid cancer are presented in Table 4.

While the data for thyroid cancer do not provide significant results, these polymorphisms are important in other types of cancer. Figueroa et al. showed an association of the heterozygous rs1805377 *XRCC4* genotype (OR = 1.43; 95% CI = 1.14–1.80; *p* = 0.002) with the risk of bladder cancer in the Spanish population. Tseng et al. reported a significant association of rs1805377 *XRCC4* in the homozygous variant (OR = 2.38; 95% CI = 1.05–5.76; *p* = 0.043) with lung cancer in the Chinese population [55,56].

### 3.5. X-ray Repair Cross-Complementing Group 5 (XRCC5)/Ku80

The *XRCC5* gene is also known as *Ku80*, and its protein product plays a key role in the NHEJ DNA repair mechanism together with the Ku70 protein (encoded by *XRCC6*). Their binding to DNA facilitates the recruitment of other proteins, which initiates a sequence of events that leads to the correct connection of broken DNA ends. Furthermore, Ku80 is a key component of the synaptic complex that prepares damaged DNA ends for processing and also promotes their reconnection by the XRCC4/LIG4 complex in the NHEJ-based DNA repair process [57]. These mechanisms are important for the maintenance of genome integrity.

*XRCC5*/*Ku80* gene polymorphisms have been investigated in the context of thyroid cancer. In the study by Gomes et al., the rs2440 polymorphism was not found to be significant in the overall Portuguese (Caucasian) population, but the logistic regression test showed significance for PTC but not for FTC (adjusted OR = 2.28; 95% CI = 1.06–4.89; *p* = 0.03) [53]. The remaining polymorphisms, that is, rs1051677, rs6941, and rs1051685, did not show statistically significant genotype differences between the study and control populations. However, when analyzing gender stratification, rs1051677 was found to be associated with an increased individual risk in the male population. This study suggests that rs2440 and rs1051677 may influence individual susceptibility to thyroid cancer, particularly in the context of PTC and female gender. However, in the Chinese population, no association between rs2440 and the risk of thyroid cancer was found [54]. The number of reports on the association between *XRCC5* polymorphisms and thyroid cancer in various populations remains insufficient. The genotypic frequencies of selected SNPs of the *XRCC5* gene in various populations and their associations with the risk of thyroid cancer are presented in Table 5. We did not find any reports on the study of *XRCC6* (Ku70) polymorphisms in the context of thyroid cancer risk. However, there are studies on other tumors [58,59,60].

### 3.6. X-ray Repair Cross-Complementing Group 7 (XRCC7)/DNA-Dependent Protein Kinase (DNA-PK)

The *XRCC7* gene encodes a DNA-dependent protein kinase catalytic subunit (DNA-PKcs), composed of 4128 amino acids. The holoenzyme consists of the Ku DNA-Binding domain and a catalytic subunit, responsible for initiating the repair process. Ku binds DSBs and uses the catalytic subunit to bind DNA ends for the removal of DNA damage with the NHEJ pathway [61,62]. Studies have confirmed that cells lacking *XRCC7* show impaired DNA repair ability after exposure to damaging factors, i.e., ionizing radiation [63]. It was suggested that thyroid cancer cells with a lower expression of DNA-PKcs are more susceptible to radiation relative to those with higher DNA-PKcs levels [64]. Additionally, the *XRCC7* polymorphisms may affect the function of the protein, which translates into the risk of cancer development. The association between *XRCC7* polymorphisms and the risk of thyroid cancer has been the subject of a few case-control studies [29,47,62].

Data investigating the association of *XRCC7* polymorphisms in the context of thyroid cancer are limited to Iranian, Saudi Arabian, and non-Hispanic white populations. In the study by Rahimi et al., the authors identify the *XRCC7* rs7830743 (Ile3434Thr) polymorphism as a potential risk factor for thyroid cancer in the Iranian population, with the risk of thyroid cancer increasing as the number of C alleles increase. Furthermore, the TC genotype showed a significantly higher risk of thyroid cancer (OR = 2.42; 95% CI = 1.55–3.81; *p* = 0.0001) compared to other genotypes [62]. In contrast, a study conducted in Saudi Arabia did not confirm a significant association between *XRCC7* rs7830743 and thyroid cancer, even after expanding the study to include subsequent trials [29]. This indicates the need for caution in generalizing the results of genetic studies between different ethnic groups and sub-populations. A 2005 study among non-Hispanic white populations found no significant association between rs7003908 (T6721G) and thyroid cancer risk [47]. The genotypic frequencies of selected SNPs of the *XRCC7* gene in various populations and their associations with the risk of thyroid cancer are presented in Table 6. Research on *XRCC7* polymorphisms should be continued in other populations to better understand the genetic impact on thyroid cancer risk.

### 3.7. TP53 Gene

The *TP53* gene, also called the guardian of the genome, plays a key role in cancer pathogenesis as a tumor suppressor gene. Its 393 amino acid protein product P53 functions as a transcription factor, regulating processes related to DNA repair, aging, cell cycle control, autophagy, and apoptosis. Research suggests that gene polymorphisms or pathogenic mutations may, on the one hand, adapt organisms to extreme environmental conditions and, on the other hand, promote the development of cancer. It is the most frequently mutated gene in human cancers, with a significant influence on the process of carcinogenesis [65]. Rs1042522 (Arg72Pro) is the most well-known *TP53* polymorphism and is believed to promote cancer because it reduces the ability of the p53 protein to activate apoptosis.

Boltze et al. were among the first to report the rs1042522 *TP53* polymorphism and susceptibility to thyroid cancer in Caucasian patients [66]. The authors observed a positive correlation between age with worse clinical prognosis and disease stage. However, the comparison of genotypes with tumor stage showed no correlation (*p* = 0.41). Interestingly, homozygous Arg/Arg (*p* < 0.01) and Pro/Pro (*p* < 0.001) genotypes showed a statistically significant incidence of poor lymph node status and distant metastases. Considering histological type, the investigators observed that the homozygous proline variant was absent in either benign thyroid adenoma or differentiated thyroid cancer (PTC and FTC). However, 100% of undifferentiated thyroid carcinoma (UTC) cases had the homozygous proline variant [66]. Granja et al. studied rs1042522 in a Brazilian population (white and non-white) in patients with PTC, FTC, and benign nodules [67]. Multivariate analyses, taking into account gender, age, smoking, and drugs, showed an association of the Pro/Pro genotype with a 5.299-fold higher risk of PTC (OR = 5.299; 95% CI = 2.334–40.436; *p* = 0.0074) and a 9.714-fold higher risk of FTC (OR = 9.714; 95% CI = 2.334–40.436; *p* = 0.0043). Similarly, Bufalo et al. confirmed that the Pro/Pro genotype variant increased the risk of PTC more than three-fold (OR = 3.522; 95% CI = 1.686–7.357; *p* = 0.0008) [68]. However, the study by Reis et al. on the Brazilian population did not show statistically significant associations for the Pro/Pro genotype [69]. The heterozygous Arg/Pro variant was significantly associated with the presence of malignant neoplastic nodules (OR = 3.65; 95% CI = 1.69–7.91; *p* = 0.001), while the homozygous Arg/Arg genotype reduced their risk (OR = 0.15; 95% CI = 0.06–0.39; *p* < 0.0001). The homozygous arginine allele may have a protective effect against carcinogenesis in the study population [69]. Yet another study did not find an association between the genetic profiles of *TP53* and susceptibility to hereditary medullary thyroid cancer (*p* = 0.102) [70]. Studies on *TP53* gene polymorphism and thyroid cancer incidence take into account not only population differences but also the influence of environmental factors. Akulevich et al., in a study on the Russian and Belarusian populations, assessed the relationship between the rs1042522 polymorphism and the risk of radiation-induced and sporadic PTC [71]. Analyzing different inheritance models, they did not observe a significant effect of polymorphism in the sporadic PTC group but noted an increased risk of radiation-induced PTC in the codominant model (OR = 2.33; 95% CI = 1.15–7.21; *p* = 0.03) and the multiplicative model (OR = 1.70; 95% CI = 1.17–2.46; *p* = 0.006) compared to radiation-exposed controls. Furthermore, analyses were performed on combinations of *ATM* and *TP53* genotypes that are functionally related. In particular, the *ATM/TP53* GG/TC/CG/GC genotype was associated with radiation-induced PTC (OR = 2.10; 95% CI = 1.17–3.78; *p* = 0.015), and the GG/CC/GG/ GG showed a significantly increased risk of sporadic PTC (OR = 3.32; 95% CI = 1.57–6.99; *p* = 0.002) [71]. Another analyzed SNP was rs17880604 (intron 6 G13964C), which in the Iranian-Azeri patients population did not show a significant association with an increased risk of developing thyroid cancer [72]. The significance of this polymorphism with any clinicopathological subtype was also not demonstrated. In summary, the studies suggest an important role of the *TP53* Arg72Pro polymorphism in the development of thyroid cancer and emphasize the need for further research to understand the molecular mechanisms and environmental influence on this disease. The genotypic frequencies of selected SNPs of the *TP53* gene in various populations and their associations with the risk of thyroid cancer are presented in Table 7.

### 3.8. RAD51 Recombinase (RAD51)

The *RAD51* gene encodes a 339 amino acid protein that is a key enzyme in the process of DNA repair through HR. RAD51 protein is an ATPase able to form helical nucleoprotein filaments on those strands and impacts the early stages of DSB recognition during HR [73]. Its overexpression, found in many cancer cases, leads to abnormalities in DNA repair. Moreover, it is involved in the development of cell resistance to radiation and drugs that can induce DSBs in cancer cells. RAD51 activity is regulated by other proteins, including BRCA1 and BRCA2. RAD51 is considered a therapeutic target for combating cancer drug resistance [73].

Studies have been conducted in several populations analyzing the association of *RAD51* gene polymorphisms (rs11852786, rs963917, rs1801321, rs304267, rs304270, and rs1801320) with the risk of thyroid cancer [29,43,44,47,54]. Guo et al. did not show a significant relationship between rs11852786 and rs963917 polymorphisms and the risk of thyroid cancer in the Chinese population [54]. Similarly, studies conducted in Saudi Arabia, Iran, and the non-Hispanic white population did not show a significant association between rs304267, rs304270, and rs1801320 polymorphisms and the risk of thyroid cancer [29,43,47]. The study by Bastos et al. on Caucasians from the Portuguese population showed a trend toward increased risk of thyroid cancer for people who are homozygous for the *RAD51* rs180132 polymorphic allele variant (adjusted OR = 1.9; 95% CI = 1.0–3.5; *p* = 0.057) [44]. However, as we wrote above (see Section 3.3), analyzing the data in terms of the number of alleles of the *RAD51* rs1801321 and *XRCC3* rs861539 variants, it was observed that the coexistence of three and four variant alleles of these genes was associated with a significantly higher risk of thyroid cancer. The genotypic frequencies of selected SNPs of the *RAD51* gene in various populations and their associations with the risk of thyroid cancer are presented in Table 8.

### 3.9. RAD52 Homolog (RAD52)

The *RAD52* gene, which encodes a 418 amino acid protein, is an important element of the mechanism of DSBR and genetic recombination. Its participation is important to maintain genomic stability and avoid mutations. RAD52 mediates DNA strand exchange and DNA-DNA interaction crucial for complementary DNA strands annealing during DSBR [74]. Changes in *RAD52* expression may affect the effectiveness of HR repair mechanisms and thus anti-cancer activity [74].

Until now, few studies have been conducted to analyze the relationship between *RAD52* polymorphisms and the risk of thyroid cancer [29,43]. In the study by Siraj et al. conducted in Saudi Arabia, two polymorphisms, rs11226 and rs4987206 (Gln221Glu), were selected, which showed high statistical significance compared to the control samples in PTC [29]. The rs4987206 polymorphism with the heterozygous CG genotype showed a statistically significant increased risk of thyroid cancer compared to the wild-type CC (OR = 15.57; 95% CI = 6.56–36.98; *p* < 0.001) and in the CG + GG vs. CC variant (OR = 17.58; 95% CI = 7.44–41.58; *p* < 0.001). Regarding rs11226, the CT genotype showed a statistically significant difference in risk assessment compared to the control group CC (OR = 1.53; 95% CI = 1.03–2.28; *p* < 0.05) and in the CT + TT variant vs. CC (OR = 1.922; 95% CI = 1.31–2.82; *p* < 0.001). Meanwhile, the study on the Iranian population did not show a significant difference concerning the *RAD52* rs11226 polymorphism [43]. Only the combined variants of the *RAD52, XRCC2*, and *XRCC3* genes showed an increased risk of thyroid cancer, as we wrote about in Section 3.2. The genotypic frequencies of selected SNPs of the *RAD52* gene in various populations and their associations with the risk of thyroid cancer are presented in Table 9. We found no studies on *RAD52* polymorphisms in Caucasians and Asians.

### 3.10. Breast Cancer Type 1 Susceptibility Protein (BRCA1)

BRCA1 is a tumor suppressor whose main function is to prevent uncontrolled cell growth by repairing DNA damage, regulating the cell cycle, and participating in the apoptosis process [75,76]. BRCA1 is also involved in the cellular response to DNA damage caused by ionizing radiation and is a key factor in preventing mutations and maintaining chromosome integrity. BRCA1 is strongly associated with cell proliferation and may be overexpressed in breast, ovarian, and thyroid cancers [75,77]. There are many polymorphisms in the *BRCA1* gene, which may be associated with cancer susceptibility. Although the functional consequences of these polymorphisms are not fully understood, they may affect the ability to repair DSBs and raise the possibility of an association between *BRCA1* genetic polymorphisms and thyroid cancer [78,79].

Xu et al. analyzed eight *BRCA1* SNPs, of which A1988G, rs799917, and rs16942 were significantly associated with a reduced risk of DTC [78]. However, after correction for repeated testing, only A1988G, located in the promoter region, showed statistical significance (adjusted OR = 0.63; 95% CI = 0.45–0.87; *p* = 0.036). Given its location, it may affect *BRCA1* transcriptional activity. The authors also found that having a combination of three or more favorable genotypes resulted in a more than 30% reduction in the risk of DTC (adjusted OR = 0.69; 95% CI = 0.50–0.95; *p* = 0.028). Wójcicka et al. conducted a study in the Polish population, which showed a significant relationship between allelic rs16941 and susceptibility to PTC (OR = 1.16; 95% CI = 1.04–1.28; *p* = 0.005) [80]. A significant association between rs16941 and PTC was detected in the recessive model (OR = 1.31; 95% CI = 1.04–1.67; *p* = 0.021).

The authors also emphasize the importance of *ATM-CHEK2-BRCA1* axis variants in the predisposition to PTC. Another study conducted by Guo et al. in the Chinese population showed no significant association between two *BRCA1* SNPs (rs12516 and rs8176318) and the risk of PTC [54]. The genotypic frequencies of selected SNPs of the *BRCA1* gene in various populations and their associations with the risk of thyroid cancer are presented in Table 10.

### 3.11. Breast Cancer Type 2 Susceptibility Protein (BRCA2)

BRCA2 is a key participant in the DSBR pathway through the HR mechanism. Its BRC motif mediates binding to RAD51, which allows it to maintain genome stability by efficiently repairing DNA damage [54]. *BRCA2* is considered a tumor suppressor gene because its participation in DNA repair limits the possibility of cancer transformation. Mutations in the gene lead to defects in DSBR, promote the accumulation of genetic damage, and increase the risk of developing cancer. Abnormal expression of BRCA2 is associated with various types of malignancies, such as breast cancer, ovarian cancer, and thyroid cancer [54,81,82]. It is suggested that polymorphisms in the *BRCA2* gene may affect the function of the protein and its expression, which also increase the risk of cancer.

Guo et al. investigated the rs15869 polymorphism within the 3′-UTR of the *BRCA2* gene for its potential role in thyroid cancer in the Chinese population [54]. The study showed that the CC genotype of this polymorphism was associated with a higher risk of PTC compared to the AA genotype. However, after a Bonferroni correction, this difference was no longer statistically significant. Additionally, the A to C substitution at rs15869 was identified to change the secondary structure of *BRCA2* mRNA and led to reduced expression of this gene. The discovery of altered mRNA secondary structure and reduced *BRCA2* expression suggests that this polymorphism may affect the function of this gene, which may have implications for the development of thyroid cancer. Further research on the mechanisms of action of this polymorphism is therefore necessary to fully understand its impact on the development of thyroid cancer. The genotypic frequencies of a selected SNP of the *BRCA2* gene in various populations and its associations with the risk of thyroid cancer are presented in Table 11.

### 3.12. Apurinic/Apyrimidinic Endonuclease (APE1)

The *APE1* gene plays an important role in repairing DNA damage through the BER repair mechanism. Its 318 amino acid protein has the N-terminal domain responsible for its role in redox signaling and the C-terminal domain involved in the endonuclease activity of the protein [83]. Its primary function is to cut out abasic sites in DNA. The redox function of APE1 plays a key role in regulating cell proliferation and survival by activating specific transcriptional activators.

Studies on the role of the rs3136820 polymorphism in the *APE1* gene in thyroid cancer conducted in the Chinese and Portuguese populations did not show a clear relationship with the risk of thyroid cancer [30,35]. No significant differences in the frequency of genotypes were observed between the study group and the control group. Moreover, the association of the rs1130409 *APE1* polymorphism with thyroid tumors and PTC after exposure to ionizing radiation was investigated in the Kazakh and Russian populations [83]. No statistically significant relationship was found for the tested polymorphism in this population. However, a study by Hu et al. found that a variant form of the APE1 enzyme may be associated with G2 cell cycle delay in response to ionizing radiation, which may contribute to radiation hypersensitivity and potentially increased risk of breast cancer [84]. The conclusion of these studies suggests that polymorphisms in the *APE1* gene may have various effects on cellular functions and responses to oxidative stress or radiation, but their direct relationship with the risk of thyroid cancer requires further research and confirmation. The genotypic frequencies of selected SNPs of the *APE1* gene in various populations and their associations with the risk of thyroid cancer are presented in Table 12.

### 3.13. Poly(ADP-Ribose) Polymerase-1 (PARP1)/ADP-Ribosyltransferase (ADPRT)

*PARP1*, also known as *ADPRT*, is a DNA repair gene encoding a 1014 amino acid nuclear protein that promotes poly(ADP-ribosylation) of many proteins, affecting their functions. It modifies chromatin proteins through ADP-ribosylation, affecting its compactness. Moreover, it specifically binds to DNA strand breaks and plays an important role in the repair through long-patch BER [85]. PARP1 plays a vital role in DNA damage detection and repair through interactions with other repair proteins such as XRCC1 [86]. Polymorphisms causing an amino acid change in the terminal catalytic domain of the enzyme may lead to reduced enzymatic activity and limited ability to interact with proteins. The associated impairment of BER repair capacity may increase cancer predisposition in carriers of these variants.

In a study by Chiang et al., the rs1136410 (Val762Ala) polymorphism showed no significant association with the risk of thyroid cancer [30]. However, the combination of an rs1136410 *PARP1* homozygote (Val/Val) with an rs1799782 *XRCC1* homozygote (Arg/Arg) resulted in a 3.18-fold increased risk of DTC (OR = 3.18; 95% CI = 1.02–9.87; *p* = 0.046) and a 9.25-fold increased risk of DTC with lymph node metastases (OR = 9.25; 95% CI = 2.07–41.4; *p* = 0.004). A study by Santos et al. also did not confirm a significant relationship between the rs1136410 polymorphism and the risk of thyroid cancer in the Portuguese population [35]. In turn, the study of genetic polymorphisms rs1136410, rs1805414, and rs1805404 in the context of thyroid cancer conducted by Bashir et al. in Pakistan showed differences in allele and genotype frequencies between cancer patients and controls [87]. The rs1136410 polymorphism showed a significantly higher frequency of CC homozygote in patients with thyroid cancer (OR = 1.30; 95% CI = 0.99–1.71; *p* = 0.05), especially in PTC (OR = 1.37; 95% CI = 1.03–1.84; *p* = 0.02). The rs1805414 polymorphism showed a lower risk of thyroid cancer in the presence of the combined CC + TC genotype (OR = 0.43; 95% CI = 0.27–0.67; *p* = 0.003), as well as a higher incidence of TC heterozygote in healthy people compared to thyroid cancer patients (OR = 0.80; 95% CI = 0.65–1.00; *p* = 0.05). Moreover, the heterozygous TC genotype was associated with a more than two-fold reduced risk of PTC (OR = 0.55; 95% CI = 0.40–0.74; *p* = 0.0001) and FTC (OR = 0.50; 95% CI = 0.30–0.82; *p* = 0.006). The rs1805404 polymorphism showed a significantly higher frequency of TT homozygote in the control group (OR = 0.63, 95% CI = 0.40–1.00; *p* = 0.05) and a higher frequency of TT homozygote in PTC (OR = 0.49; 95% CI = 0.28–0.83; *p* = 0.008), and anaplastic carcinoma (OR = 9.77; 95% CI = 2.12–45.0; *p* = 0.003) [87]. Therefore, polymorphisms of these genes may influence the risk of thyroid cancer, especially in the context of different histological subtypes.

In a meta-analysis, Li et al. collected research results on *PARP1* polymorphisms [88]. The authors showed that the rs1136410 polymorphism has a significant association with the overall risk of cancer, and detailed subgroup analyses showed that it may significantly predispose to stomach cancer, thyroid cancer, and cervical cancer while protecting against brain cancer. The genotypic frequencies of selected SNPs of the *PARP1* gene in various populations and their associations with the risk of thyroid cancer are presented in Table 13.

### 3.14. 8-Oxoguanine-DNA Glycosylase 1 (OGG1)

*OGG1* is a gene encoding the enzyme 8-oxoguanine-DNA glycosylase 1, responsible for the removal of 8-oxoguanine (8-oxoG), a mutagenic product resulting from exposure to reactive oxygen species. OGG1 plays a key role in DNA repair through the BER pathway, where it prevents the accumulation of genetic mutations and maintains genome stability [89]. Abnormalities in the functioning of this enzyme may lead to increased DNA damage and susceptibility to cancer development. For example, hypermethylation of *OGG1* genes was observed in papillary thyroid cancer [90].

Studies on the role of the rs1052133 (Ser326Cys) polymorphism in the *OGG1* gene conducted by García-Quispes and Santos in Spanish and Portuguese populations did not show a significant impact of this polymorphism on the risk of thyroid cancer [35,36]. However, it has been reported that polymorphisms of this gene may play a role as a modulator of DNA damage. The rs1052133 causes a slow DNA repair capacity in vitro, as observed in cells of people exposed to low doses of ionizing radiation [91]. Gene-gene interaction analysis did not reveal significant associations between *OGG1* and *XRCC1* in the context of DNA repair. The rs1052133 polymorphism in the *OGG1* gene may therefore have a limited impact on the risk of thyroid cancer, but its role in the DNA repair pathway requires further research. The genotypic frequencies of selected SNPs of the *OGG1* gene in various populations and their associations with the risk of thyroid cancer are presented in Table 14.

### 3.15. MutY DNA Glycosylase (MUTYH)

*MUTYH* plays an important role in the repair of oxidative DNA damage through the BER repair pathway. This gene encodes a 546-amino acid protein whose main function is to remove mispaired adenine residues that may arise from the presence of a stable guanine oxidation product, such as 8-oxo-7,8-dihydro-2′-deoxyguanosine (8-oxo-dG) [92]. *MUTYH* dysfunction may be particularly problematic because it lacks other mechanisms to repair 8-oxo-dG/adenine mismatches, which may contribute to human carcinogenesis. Hence, the gene and the protein it encodes play an important role in maintaining genome stability and preventing the development of cancer.

In a study by Santos et al. in the Portuguese population, there was no clear relationship between the rs3219489 (Gln335His) polymorphism in the *MUTYH* gene and the overall risk of thyroid cancer [35]. However, analysis considering only the PTC suggested that heterozygous (Gln/His) individuals may have a borderline reduction in risk of this cancer type. The adjusted OR values for the rs3219489 polymorphism were close to achieving statistical significance (adjusted OR = 0.62; 95% CI = 0.36–1.07; *p* = 0.080), suggesting a potential impact of this polymorphism on the risk of PTC. The genotypic frequencies of a selected SNP of the *MUTYH* gene in the study population and its associations with the risk of thyroid cancer are presented in Table 15.

### 3.16. Ataxia Telangiectasia Mutated (ATM)

The *ATM* gene encodes a serine/threonine protein kinase belonging to the phosphoinositide 3-kinase (PIKK)-related protein kinase family [93]. The ATM protein plays an important role in the detection and repair of DNA DSBs and in cellular responses to DNA damage, which contributes to the maintenance of genome stability. It is a central kinase responsible for activating a complex network response through signal transduction. Moreover, it phosphorylates proteins involved in histone ubiquitination and chromatin remodeling, facilitating the repair process [94]. Mutations in the *ATM* gene, located in the chromosomal region 11q22-23, may lead to conformational changes, inactivation of the kinase domain, or changes in the stability of the molecule, which may weaken the function of this gene and increase the risk of cancers. ATM is a key factor in the mechanism of DNA repair induced by ionizing radiation, where it activates the checkpoint kinase (CHK2) and BRCA1-dependent repair pathway, which can lead to DNA repair or cellular apoptosis in the event of repair failure [80]. Polymorphisms and mutations in the genes involved in the *ATM-BRCA1-CHK2* pathway can lead to inefficient DNA repair and predispose to cancer. This highlights the importance of *ATM* as a factor regulating genome stability and the cellular response to DNA damage, as well as its potential role in cancer pathogenesis.

Studies on *ATM* gene polymorphisms in the context of thyroid cancer have shown different results depending on the population studied. In a study conducted by Wójcicka et al. among the Polish population, it was found that rs1801516 (Asp1853Asn) polymorphism was not associated with the risk of thyroid cancer but played a role as a risk modifier associated with other genes, such as *BRCA1*, mitigating the negative impact of the rare BRCA1 variant by 0.78 (*p* = 0.023) [80]. In a study in China by Gu et al., of the four *ATM* gene polymorphisms tested (rs664677, rs4988099, rs189037, rs373759), only the rs373759 polymorphism showed a significant association with the risk of PTC. The AG genotype of this polymorphism was significantly associated with a 1.38-fold increased risk of PTC (adjusted OR = 1.38; 95% CI = 1.03–1.87; *p* = 0.03) [95]. Another study on the Chinese population investigated the association of *ATM* polymorphisms with gender-specific PTC metastases [96]. An association was found between the rs189037 polymorphism and PTC metastases in women in various inheritance models. However, after a Bonferroni correction, no relationship remained statistically significant. The analysis by Xu et al. in the non-Hispanic white population proved that polymorphisms rs189037 and rs1800057 may also influence the risk of thyroid cancer [97]. In the dominant inheritance model, rs189037, the G allele of *ATM* showed a protective effect against DTC (adjusted OR = 0.8; 95% CI = 0.6–1.0; *p* = 0.04). However, the G allele for rs1800057 was associated with an increased risk of DTC (adjusted OR = 1.9; 95% CI = 1.1–3.1; *p* = 0.02). Additionally, a dose-response relationship was observed between the number of *ATM*-risk alleles and the risk of thyroid cancer (*p* = 0.01). A study conducted among the Polynesian population, which is a geographically isolated group exposed to ionizing radiation, showed a different impact of the rs1801516 polymorphism on the risk of thyroid cancer compared to other populations [98]. The differences were primarily due to the frequency of the rs1801516 alleles. The A allele of rs1801516 was associated with an increased risk of thyroid cancer among Polynesians (OR = 3.13; 95% CI = 1.17–8.31; *p* = 0.02), which is opposite to that of the European population. In a Cuban population, Pereda et al. did not observe an association between DTC and rs1801516 *ATM* (OR = 1.2; 95% CI = 0.7–2.0; *p* = 0.8) [99]. Interestingly, in women who had two or more pregnancies, the risk of DTC was 3.5-fold higher if they were carriers of the A allele (OR = 3.5; 95% CI = 3.2–9.8; *p* = 0.03) compared to 0.8 (OR = 0.8; 95% CI = 0.4–1.6) in women with less than two pregnancies. Akulevich et al. studied a population of Russians and Belarusians, some of whom lived in areas exposed to the Chernobyl explosion, to investigate the relationship between genetic polymorphisms and susceptibility to the disease [71]. A total of 255 cases of PTC were examined, including 123 caused by Chernobyl radiation, 132 sporadic cases, and 596 healthy controls. The study included people who were under 18 years of age at the time of the Chernobyl accident, with individual thyroid radiation doses. An association was found between rs1801516 and PTC, regardless of radiation exposure. The presence of the A allele significantly reduced the risk of PTC compared to the wild-type G allele in the multiplicative model of inheritance (OR = 0.69; 95% CI = 0.45–0.86; *p* = 0.03). Furthermore, there was an interaction between radiation exposure and PTC for rs664677 *ATM* (*p* = 0.04). Analyses performed in patients exposed to ionizing radiation and not irradiated showed an increased risk of sporadic PTC in carriers of the homozygous CC rs664677*ATM* genotype compared to TC + TT genotypes (OR = 1.84; 95% CI = 1.10–3.24; *p* = 0.03). There was no significant effect of rs609429 on the risk of thyroid cancer. However, the study by Damiola et al. on the population of Belarusian children exposed to ionizing radiation due to the Chernobyl nuclear accident showed a reduced risk of PTC in carriers of the *ATM* allele A of rs1801516 in relation to the risk of PTC (OR = 0.34; 95% CI = 0.16–0.73, *p* = 0.0017) [100]. A meta-analysis summarizing the importance of *ATM* polymorphisms and the risk of thyroid cancer was found. However, after detailed analyses, the authors did not find any positive associations between *ATM* genetic polymorphisms and PTC (*p* > 0.05) [101]. The genotypic frequencies of selected SNPs of the *ATM* gene in various populations and their associations with the risk of thyroid cancer are presented in Table 16.

### 3.17. Checkpoint Kinase 2 (CHK2)

The *CHK2* gene encodes the CHK2 kinase that plays a key role in the cellular response to DNA damage and the regulation of the cell cycle through phosphorylation of CDC25 phosphatases [102]. It is activated in response to DSB emergence, ionizing radiation, or other genotoxic factors. It initiates a signaling cascade that leads to cell cycle arrest, activation of DNA repair processes, or cell death, thus ensuring proper control over cellular proliferation and genome integrity.

The research carried out by Wójcicka et al. in the Polish population showed a significant increase in the risk of PTC associated with the rs17879961 (Ile157Thr) polymorphism in the *CHK2* gene in the dominant model (OR = 2.21; 95% CI = 1.69–2.88; *p* = 9.60 × 10^−10^) [80]. This confirmed the suspicion that this variant leads to the production of a protein with limited functionality, which may cause disruptions in cellular processes and an increased risk of cancer. According to some researchers, the rs17879961 polymorphism may be a founding mutation in various ethnic populations. Polymorphisms in the *CHK2* gene have also been analyzed in patients with other cancers, including breast cancer, and a variant was identified that was more common in cancer patients than in healthy people, suggesting that *CHK2* polymorphisms may be associated with cancer risk [102]. We found no studies examining *CHK2* gene polymorphisms in other populations. The genotypic frequencies of a selected SNP of the *CHK2* gene in the study population and its associations with the risk of thyroid cancer are presented in Table 17.

## 4. Concluding Remarks

Thyroid cancer is one of the most common forms of endocrine cancer and one of the fastest-growing cancer diseases in the world. An increase in the incidence of thyroid cancer has been observed in underdeveloped and highly developed countries, where routine tests and diagnostics are common. The disease develops in tissues of the thyroid gland, which is responsible for the production of hormones that regulate metabolism. Thyroid cancer can come in many forms, but the most common are PTC and FTC. PTC, which accounts for approximately 80–85% of all cases, is characterized by the rapid growth of cancer cells that form wart-like structures [14]. It is prone to metastasize to lymph nodes and distant organs. It can occur in different varieties due to morphology and focality [15,103]. However, FTC, which accounts for approximately 5–15% of all cases of thyroid cancer, is characterized by the formation of a follicle-like structure. This type of cancer is less likely to spread to the lymph nodes than PTC but can grow to a large size. FTC is often intermediate between benign and malignant tumors, which means it does not always have the aggressive features typical of malignant tumors [103].

Thyroid cancer may arise from a range of risk factors, some of which differ among populations. One of the main factors contributing to morbidity is exposure to ionizing radiation. Ionizing radiation damages thyroid cells, leading to uncontrolled cell growth and cancer transformation. People who receive head and neck radiotherapy for other conditions at a young age are at increased risk of developing thyroid cancer in the future [104]. Additionally, nuclear disasters such as the Chernobyl and Fukushima accidents resulted in the release of large amounts of radioactive substances into the environment. People exposed to these substances, especially radioactive iodine, have an increased risk of developing thyroid cancer. As a result of the Chernobyl disaster, there was a significant increase in thyroid cancer cases among children and adolescents in affected regions [100,105]. However, the onset of thyroid cancer after radiation exposure can take several decades. Other risk factors include iodine deficiency in the diet. Iodine is an element necessary for hormone production, and deficiency can lead to the formation of thyroid tumors [106]. Another predisposing factor is gender. Gender differences in disease incidence, course, aggressiveness, and prognosis have been documented [107,108]. Furthermore, this risk increases with age, especially in people over 45–50 years of age. The last key factor is genetic predisposition. As with other tumor types, people with a family history of thyroid cancer have an increased risk of developing the disease. The familial tendency results from the existence of genetic factors, i.e., genetic mutations and gene polymorphisms.

Gene polymorphisms, especially DNA repair genes, play an important role in carcinogenesis by influencing signal transduction and genome stability. Polymorphisms may affect the function of proteins involved in DNA repair by changing their structure or enzymatic activity, which may lead to a reduction or increase in the effectiveness of DNA damage repair systems, including BER, NER, MMR, HR, and NHEJ. People with polymorphisms that reduce the function of DNA repair proteins may be more susceptible to the accumulation of mutations in thyroid cells and at an increased risk of carcinogenesis. Moreover, polymorphisms in DNA repair genes may influence the response to anticancer therapy, including the effectiveness of chemotherapy and radiotherapy, which is of significant clinical importance in the context of selecting appropriate therapeutic strategies and predicting treatment outcomes. Therefore, studies aimed at identifying polymorphisms in DNA repair genes may be an important tool in assessing the risk of developing thyroid cancer and in adapting therapeutic strategies to the individual needs of patients. In this review, we collected and discussed polymorphisms of selected genes related to DNA repair. The work does not constitute a meta-analysis but is only a literature summary of existing works. The lack of references for some populations is due to the lack of available reports on them. These constitute an important research gap.

Variations in genetic polymorphisms can have a significant impact on the risk of developing thyroid cancer in different populations. Research collected in this work indicates the existence of significant differences in the frequency of specific polymorphisms in DNA repair genes and the relationship with thyroid cancer between different ethnic groups. For example, the TT genotype of the rs1799782 polymorphism in the *XRCC1* gene has been associated with an increased risk of thyroid cancer in all analyzed Chinese populations and in the non-Hispanic white population but with a decreased risk in the Pakistani population [27,30,31,32,37]. The AA genotype of the rs25489 polymorphism in the *XRCC1* gene significantly reduced the risk of thyroid disease in the Pakistani population, while in the heterozygote, it was associated with increased DTC in the Spanish population [27,28,36]. Similarly, Caucasian populations have been observed to have a higher risk associated with certain polymorphisms compared to Asian populations. The impact of these polymorphisms on the risk of thyroid cancer may be modulated by environmental factors, such as exposure to ionizing radiation, which is particularly important in the case of populations that were repeatedly exposed to this type of radiation, such as the Polynesian or Belarusian population [71,98,100,105]. Other external factors that can lead to polymorphisms that alter the regulation of DNA repair genes include chemicals present in tobacco smoke, some air pollutants, pesticides, and other environmental toxins that can induce mutations in DNA repair genes [109,110,111,112]. In addition, some dietary components, including substances found in processed or contaminated foods, can cause DNA damage and lead to polymorphisms in DNA repair genes [113,114]. Similarly, some viruses such as human papillomavirus and others can introduce changes to DNA [115]. Other factors include drugs used, exposure to high temperatures, oxidative stress, genotoxins, and mechanical damage to cells [116]. Understanding the diversity of genetic polymorphisms in different populations and their interactions with environmental factors may contribute to a better understanding of the pathophysiological mechanisms of thyroid cancer and lead to more effective prevention and treatment strategies for this disease, especially in populations at increased risk.

Low-fidelity polymerases, which play a crucial role in DNA repair processes, are among the other factors involved in the development of cancer, including thyroid cancer.

On the one hand, polymorphisms in DNA repair genes may affect the expression and functionality of these polymerases. On the other hand, their activity affects genetic variability and the ability of cancer cells to adapt to anticancer therapies. Low-fidelity polymerases are particularly important in the context of therapeutic resistance because their action may lead to the formation of mutations that favor the survival of cancer cells despite therapy [117]. EGFR and AXL receptors play a significant role in regulating the expression of DNA repair genes and high- and low-fidelity polymerases [118,119]. EGFR may down-regulate DNA repair genes from the MMR and HR systems, increasing the likelihood of polymerase errors [120]. Activation of these receptors can influence various signaling pathways that modulate the cell response to DNA damage, increasing the ability of cancer cells to survive in the face of targeted therapy [121]. Low-fidelity polymerases can compensate for the loss of function of high-fidelity polymerases, which contributes to the adaptive resistance of cancer cells. Understanding these mechanisms is the key to developing more effective treatment strategies that could counter cell resistance by modulating the activity of polymerases and receptors involved in DNA repair. There are reports on proteins such as REV1, indicating their potential role as biomarkers in the evaluation of the prognosis and effectiveness of therapy [122]. Analysis of REV1 expression and its polymorphisms can provide valuable information on predicting response to treatment, as well as the risk of developing resistance. Furthermore, understanding the regulation of these polymerases and their interactions with high-fidelity polymerases may help develop more precise therapeutic strategies to minimize the risk of disease recurrence and progression. Unfortunately, we did not find studies that examined polymorphisms in DNA repair genes that would translate into the functionality of low-fidelity polymerases. This constitutes a very interesting research gap.

Polymorphism of genes such as *RAD52*, *BRCA2*, *APE1*, *OGG1*, *MUTYH*, and *CHK2* were analyzed only in one, two, or three populations limited to country or ethnic group. The data collected in this way do not fully illustrate the impact of polymorphisms of these genes on the incidence of thyroid cancer. The existence of research gaps in the context of such limited results highlights the need for continued research to obtain a more complete picture in all possible population contexts. Moreover, we did not find information on testing polymorphisms of other DNA repair genes, i.e., *XRCC6*, *CHK1*, *XPA*, *ERCC3/XPB*, *NEIL-2*, *NTH-1*, and *FEN-1*, although they were tested in other types of cancer. While *CHK2* polymorphisms have been extensively studied in thyroid cancer, there is a lack of reports on *CHK1* polymorphisms, which have been studied in nasopharyngeal cancer (NPC) and ovarian cancer [123,124]. The rs492510 *CHK1* polymorphism increased the transcriptional activity of the CHK1 protein along with functionality, contributing to increased susceptibility to NPCs. In the case of the rs1800975 polymorphism, *XPA* has been shown to reduce the risk of breast cancer and esophageal cancer in the Asian population [125,126]. Polymorphisms *ERCC3/XPB*, *NEIL-2*, *NTH-1*, and *FEN-1* have been studied in lung, gastric, lung, and breast cancer [127,128,129,130,131,132,133]. The lack of reports on their role in thyroid cancer in various populations sets a new niche in the field, thus providing a potential opportunity for other researchers to explore undiscovered areas related to this disease.

## Figures and Tables

**Figure 1 ijms-25-05995-f001:**
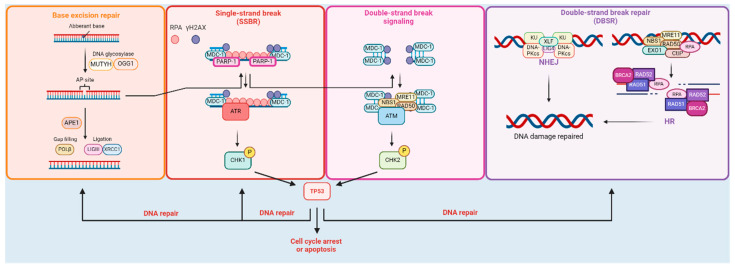
Summary of DNA repair mechanisms described in this manuscript. The base excision repair (BER) pathway involves the repair of damaged bases that are cleaved by DNA glycosylases, i.e., 8-oxoguanine-DNA glycosylase 1 (OGG1) and adenine DNA glycosylase (MUTYH), leading to the formation of apurinic/apyrimidinic sites (AP sites), which constitute substrate for DNA nuclease (APE1). Then, polymerase beta (POLβ) synthesizes the missing nucleotide, and ligase III (LIGIII) in a complex with XRCC1 restores the missing phosphodiester bond. AP sites are easily converted to single-strand breaks (SSBs), which are detected by poly(ADP-ribose) polymerase-1 (PARP-1). The damage site is marked by Ser139-phosphorylated H2AX (γH2AX), replication protein A (RPA), and the mediator of DNA damage checkpoint protein 1 (MDC-1), resulting in the recruitment of ATR kinase and phosphorylation of checkpoint kinase 1 (CHK1). Unrepaired SSBs can be converted to double-strand breaks (DSBs), which are detected by the MRN complex, composed of meiotic recombination 11/radiation sensitive 50/nibrin proteins (MRE11/RAD50/NBS1). The signal requires further amplification and transduction using the ataxia telangiectasia mutated (ATM) transducer protein kinase, which, as a result of interaction with the C-terminus of NBS1, becomes autophosphorylated and localizes to the site of damage. Checkpoint kinase 2 (CHK2) constitutes one of the targets of ATM kinase. Phosphorylated CHK1 and CHK2 kinases interact with the cellular tumor antigen p53 (TP53), which acts as a transcription regulator and determines the choice between DNA repair, cell cycle arrest, and apoptosis. TP53 can interact with APE1 and POLβ in the BER pathway. Repair of DSBs can be carried out by non-homologous end joining (NHEJ) or homologous recombination (HR). In the case of NHEJ, DNA-dependent protein kinase catalytic subunits (DNA-PKcs) and KU proteins are expressed, which combine with non-homologous end-joining factor 1 (XLF) and ligase 4 (LIG4) to form a repair complex. Meanwhile, in HR repair, the MRN complex recruits CtBP-interacting protein (CtIP), which binds to the MRN complex and causes activation of the endonucleolytic properties of Mre11. Mre11 works as an exonuclease that triggers degradation of the fragment, which results in the formation of a short single-stranded DNA fragment, and exonuclease 1 (EXO1) further degrades the fragment in the 3′-->5′ direction. On the side of the DNA break, the single-stranded fragment is covered with RPA proteins. Breast cancer type 2 susceptibility protein 2 (BRCA2) is recruited, which co-localizes with the DNA repair protein RAD51 homolog 1 (RAD51) and promotes homology search and strand invasion, resulting in DNA damage repair. Created with BioRender.com (accessed on 28 February 2024).

**Table 1 ijms-25-05995-t001:** Genotypic frequencies of selected SNPs of the *XRCC1* gene in different populations and their associations with the risk of thyroid cancer.

Population	SNP Reference (rs)	Genotype	OR (95% CI)	*p*-Value	References
Pakistani	rs25489	GG	1		[27]
		GA	1.05 (0.74–1.47)	0.77 (n.s.)	
		**AA**	**0.17 (0.10–0.31)**	**0.0001**	
		**GA + AA**	**0.53 (0.39–0.72)**	**0.0001**	
Chinese	rs25489	GG	1		[32]
		GA	1.10 (0.89–1.53)	0.55 (n.s.)	
		AA	1.20 (0.72–1.98)	0.45 (n.s.)	
		GA + AA	1.13 (0.84.153)	0.40 (n.s.)	
Saudi Arabian	rs25489	GG	1		[29]
		GA	0.59 (0.29–1.22)	0.173 (n.s.)	
		AA	0.93 (0.33–2.65)	1.0 (n.s.)	
		GA + AA	0.67 (0.35–1.26)	0.268 (n.s.)	
Spanish	rs25489	GG	1		[36]
		GA	1.61 (1.05–2.46)	0.087 (n.s.)	
		AA	1.22 (0.24–6.24)		
		**GG vs. AA + GA**	**1.58 (1.05–2.40)**	**0.027**	
Chinese	rs25489	GG	1		[31]
		GA	0.73 (0.50–1.07)	0.13 (n.s.)	
		AA	1.02 (0.35–2.99)	1.0 (n.s.)	
		GA + AA	0.75 (0.52–1.08)	0.144 (n.s.)	
Chinese	rs25489	GG	1		[30]
		GA	0.74 (0.51–1.07)	n.s.	
		AA	1.07 (0.33–3.41)	n.s.	
Chinese	rs25487	GG	1		[32]
		GA	1.07 (0.77–1.48)	0.66 (n.s.)	
		AA	1.20 (0.72–1.98)	0.45 (n.s.)	
		GA + AA	1.10 (0.81–1.48)	0.52 (n.s.)	
Saudi Arabian	rs25487	GG	1		[29]
		GA	0.73 (0.37–1.47)	0.495 (n.s.)	
		AA	0.54 (0.12–2.48)	0.536 (n.s.)	
		GA + AA	0.70 (0.36–1.36)	0.333 (n.s.)	
Non-Hispanic white	rs25487	GG	1		[37]
		GA	0.8 (0.6–1.0)	n.s.	
		**AA**	**0.5 (0.3–0.8)**	**0.007**	
		**GA or AA**	**0.7 (0.5–1.0)**	**0.017**	
Korean	rs25487	GG	1		[33]
		GA	0.74 (0.359–1.529)	0.51 (n.s.)	
		AA	0.64 (0.228–1.814)	0.405 (n.s.)	
Portuguese	rs25487	GG	1		[35]
		GA	0.90 (0.55–1.47)	n.s.	
		AA	0.98 (0.46–2.14)	n.s.	
Chinese	rs25487	GG	1		[31]
		GA	0.75 (0.54–1.04)	0.0983 (n.s.)	
		**AA**	**0.49 (0.28–0.84)**	**0.0098**	
		**GA + AA**	**0.69 (0.50–0.93)**	**0.0191**	
Chinese	rs25487	GG	1		[30]
		GA	1.25 (0.91–1.71)	n.s.	
		AA	1.58 (0.87–2.86)	n.s.	
Pakistani	rs1799782	CC	1		[27]
		CT	0.88 (0.66–1.16)	0.38 (n.s.)	
		**TT**	**0.71 (0.50–1.01)**	**0.05**	
		**CT + TT**	**0.55 (0.38–0.81)**	**0.002**	
Chinese	rs1799782	CC	1		[30]
		CT	1.08 (0.79–1.48)	0.614 (n.s.)	
		**TT**	**1.85 (1.11–3.07)**	**0.018**	
Chinese	rs1799782	CC	1		[32]
		CT	1.24 (0.83–1.85)	0.26 (n.s.)	
		**TT**	**2.12 (1.32–3.41)**	***p* < 0.05**	
		**CT + TT**	**1.53 (1.10–2.12)**	***p* < 0.05**	
Non-Hispanic whites (Teksas)	rs1799782	CC	1		[37]
		CT	1.40 (0.90–2.10)	n.s.	
		**TT**	**10.4 (1.0–105.5)**	**0.048**	
		CT or TT	1.50 (1.00–2.30)	n.s.	
Korean	rs1799782	CC	1		[33]
		**CT**	**0.55 (0.308–0.983)**	**0.044**	
		TT	0.403 (0.159–1.025)	0.056 (n.s.)	
Portuguese	rs1799782	CC	1		[35]
		CT	0.76 (0.44–1.78)	n.s.	
		TT	n.d.	n.d.	
Chinese	rs1799782	CC	1		[31]
		CT	1.118 (0.80–1.54)	0.508 (n.s.)	
		**TT**	**2.327 (1.36–3.96)**	**0.002**	
		CT + TT	1.232 (0.90–1.67)	0.185 (n.s.)	

Results in **bold** are statistically significant. n.s.—no significance. n.d.—not determined.

**Table 2 ijms-25-05995-t002:** Genotypic frequencies of selected SNPs of the *XRCC2* gene in different populations and their associations with the risk of thyroid cancer.

Population	SNP Reference (rs)	Genotype	OR (95% CI)	*p*-Value	References
Spanish	rs3218536	GG	1		[36]
		GA	1.12 (0.80–1.59)	n.s.	
		AA	1.36 (0.33–5.63)	n.s.	
Portuguese	rs3218536	GG	1		[44]
		GA	0.8 (0.4–1.6)	n.s.	
Chinese	rs3218536	GG	1		[31]
		GA	1.076 (0.73–1.58)	0.766 (n.s.)	
		AA	1.486 (0.29–7.43)	0.689 (n.s.)	
		AA + GA	1.91 (0.74–1.59)	0.697 (n.s.)	

n.s.—no significance.

**Table 3 ijms-25-05995-t003:** Genotypic frequencies of selected SNPs of the *XRCC3* gene in different populations and their associations with the risk of thyroid cancer.

Population	SNP Reference (rs)	Genotype	OR (95% CI)	*p*-Value	References
Spanish	rs1799796	AA	1		[36]
		AG	0.94 (0.71–1.25)	n.s.	
		GG	0.63 (0.34–1.18)	n.s.	
Chinese	rs1799796	AA	1		[31]
		AG	1.113 (0.80–1.53)	0.564 (n.s.)	
		GG	1.364 (0.77–2.38)	0.312 (n.s.)	
		AG + GG	1.154 (0.84–1.56)	0.390 (n.s.)	
Portuguese	rs861539	TT	1		[44]
		TM	0.7 (0.4–1.2)	n.s.	
		**MM**	**2.0 (1.1–3.6)**	**0.026**	
Chinese	rs861539	CC	1		[31]
		TC	1.373 (0.98–1.91)	0.071 (n.s.)	
		**TT**	**2.918 (1.65–5.15)**	**0.0003**	
		**CT + TT**	**1.602 (1.17–2.18)**	**0.0033**	
non-Hispanic white	rs861539	CC	1		[47]
		**TC**	**2.1 (1.2–3.5)**	**0.006**	
		TT	2.1 (1.0–4.4)	0.055 (n.s.)	
		**CT or TT**	**2.1 (1.3–3.4)**	**0.002**	
Saudi Arabian	rs861539	CC	1		[29]
		CT	0.62 (0.28–1.35)	0.245 (n.s.)	
		TT	1.51 (0.57–4.01)	0.429 (n.s.)	
		CT + TT	0.79 (0.39–1.58)	0.592 (n.s.)	
Chinese	rs1799794	AA	1		[31]
		AG	1.374 (0.99–1.90)	0.057 (n.s.)	
		GG	1.437 (0.85–2.40)	0.182 (n.s.)	
		**AG + GG**	**1.386 (1.01–1.88)**	**0.042**	
Chinese	rs56377012	AA	1		[31]
		AG	0.940 (0.57–1.53)	0.564 (n.s.)	
		GG	2.360 (0.76–7.30)	0.156 (n.s.)	
		AG + GG	1.078 (0.68–1.69)	0.818 (n.s.)	

Results in **bold** are statistically significant. n.s.—no significance.

**Table 4 ijms-25-05995-t004:** Genotypic frequencies of selected SNPs of the *XRCC4* gene in different populations and their associations with the risk of thyroid cancer.

Population	SNP Reference (rs)	Genotype	OR (95% CI)	*p*-Value	References
Portuguese	rs1805377	GG	1		[53]
		GA	0.6 (0.31–1.14)	n.s.	
		AA	0.31 (0.04–2.53)	n.s.	
Saudi Arabian	rs1805377	GG	1		[29]
		GA	0.57 (0.28–1.14)	0.134 (n.s.)	
		AA	0.64 (0.14–3.01)	0.738 (n.s.)	
		GA + AA	0.58 (0.30–1.12)	0.11 (n.s.)	
Chinese	rs2035990	CC	1		[35]
		CT	0.811 (0.511–1.285)	0.371 (n.s.)	
		TT	0.854 (0.501–1.457)	0.562 (n.s.)	
Portuguese	rs28360135	GG	1		[53]
		GA	1.24 (0.59–2.59)	n.s.	
		AA	n.d.	n.d.	

n.s.—no significance. n.d.—not determined.

**Table 5 ijms-25-05995-t005:** Genotypic frequencies of selected SNPs of the *XRCC5* gene in different populations and their associations with the risk of thyroid cancer.

Population	SNP Reference (rs)	Genotype	OR (95% CI)	*p*-Value	References
Portuguese	rs2440	GG	1		[53]
		GA	1.34 (0.78–2.28)	n.s.	
		AA	1.56 (0.77–3.17)	n.s.	
Chinese	rs2440	AA	1		[35]
		AG	0.911 (0.605–1.373)	0.657 (n.s.)	
		GG	0.797 (0.390–1.631)	0.535 (n.s.)	
Portuguese	rs1051677	TT	1		[53]
		TC	1.28 (0.29–4.36)	n.s.	
		CC	0.70 (0.07–6.98)	n.s.	
Portuguese	rs6941	CC	1		[53]
		CA	1.19 (0.68–2.07)	n.s.	
		AA	0.70 (0.70–7.01)	n.s.	
Portuguese	rs1051685	AA	1		[53]
		AG	1.19 (0.68–2.07)	n.s.	
		GG	0.70 (0.07–7.01)	n.s.	

n.s.—no significance.

**Table 6 ijms-25-05995-t006:** Genotypic frequencies of selected SNPs of the *XRCC7* gene in different populations and their associations with the risk of thyroid cancer.

Population	SNP Reference (rs)	Genotype	OR (95% CI)	*p*-Value	References
Iranian	rs7830743	CC vs. TT	1.16 (0.25–5.28)	0.8490 (n.s.)	[62]
		**TC vs. TT**	**2.42 (1.55–3.81)**	**0.0001**	
		**CC or TC vs. TT**	**2.32 (1.49–3.61)**	**0.0002**	
		CC vs. TC or TT	0.88 (0.19–4.00)	0.8710 (n.s.)	
Saudi Arabian	rs7830743	AA	1		[29]
		AG	1.30 (0.57–2.95)	0.515 (n.s.)	
		GG	4.97 (1.06–23.4)	0.060 (n.s.)	
		AG + GG	1.59 (0.75–3.36)	0.223 (n.s.)	
Non-Hispanic white	rs7003908	TT	1		[47]
		TG	1.2 (0.7–2.0)	n.s.	
		GG	1.0 (0.5–2.2)	n.s.	
		TG or GG	1.2 (0.7–1.9)	n.s.	

Results in **bold** are statistically significant. n.s.—no significance.

**Table 7 ijms-25-05995-t007:** Genotypic frequencies of selected SNPs of the *TP53* gene in different populations and their associations with the risk of thyroid cancer.

Population	SNP Reference (rs)	Genotype	OR (95% CI)	*p*-Value	References
Russian and Belarus	rs1042522	GG	1		[71]
		GC	1.02 (0.70–1.47)	0.89 (n.s.)	
		CC	1.16 (0.63–2.14)	0.38 (n.s.)	
Brazilians	rs1042522	**GG**	**0.15 (0.06–0.39)**	***p* < 0.0001**	[69]
		**GC**	**3.65 (1.69–7.91)**	**0.001**	
		CC	1.91 (0.75–4.85)	0.26 (n.s.)	
Iranian-Azeri	rs17880604	GG	1		[72]
		GC	0.583 (0.126–2.313)	0.564 (n.s.)	
		CC	n.d.	n.d.	

Results in **bold** are statistically significant. n.s.—no significance. n.d.—not determined.

**Table 8 ijms-25-05995-t008:** Genotypic frequencies of selected SNPs of the *RAD51* gene in different populations and their associations with the risk of thyroid cancer.

Population	SNP Reference (rs)	Genotype	OR (95% CI)	*p*-Value	References
Chinese	rs11852786	CC	1		[54]
		CG	1.491 (0.971–2.288)	0.068 (n.s.)	
		GG	2.567 (0.869–7.585)	0.088 (n.s.)	
non-Hispanic white	rs1801320	GG	1		[47]
		GC or CC	1.5 (0.7–3.0)	n.s.	
Iranian	rs1801320	GG	1		[43]
		GC	0.95 (0.54–1.67)	0.84 (n.s.)	
Chinese	rs963917	AA	1		[54]
		AG	1.179 (0.748–1.859)	0.479 (n.s.)	
		GG	0.933 (0.532–1.634)	0.808 (n.s.)	
Portuguese	rs1801321	GG	1		[44]
		GT	1.5 (0.8–2.5)	n.s.	
		TT	1.9 (1.0–3.5)	0.057 (n.s.)	
Saudi Arabian	rs304267	TT	1		[29]
		TC	0.89 (0.46–1.72)	0.741 (n.s.)	
		CC	0.54 (0.19–1.54)	0.337 (n.s.)	
		TC + CC	0.80 (0.43–1.50)	0.519 (n.s.)	
Saudi Arabian	rs304270	CC	1		[29]
		CT	1.63 (0.81–3.27)	0.174 (n.s.)	
		TT	2.55 (0.94–6.91)	0.084 (n.s.)	
		CT + TT	1.78 (0.91–3.46)	0.107 (n.s.)	

n.s.—no significance.

**Table 9 ijms-25-05995-t009:** Genotypic frequencies of selected SNPs of the *RAD52* gene in different populations and their associations with the risk of thyroid cancer.

Population	SNP Reference (rs)	Genotype	OR (95% CI)	*p*-Value	References
Saudi Arabian	rs11226	CC	1		[29]
		**CT**	**1.53 (1.03–2.28)**	**0.036**	
		CT	n.d.	n.d.	
		**CT + TT**	**1.92 (1.31–2.82)**	***p* < 0.001**	
Iranian	rs11226	CC	1		[43]
		CT + TT	1.04 (0.69–1.59)	0.83 (n.s.)	
Saudi Arabian	rs4987206	CC	1		[29]
		**CG**	**15.57 (6.56–36.98)**	***p* < 0.001**	
		GG	n.d.	n.d.	
		**CG + GG**	**17.58 (7.44–41.58)**	***p* < 0.001**	

Results in **bold** are statistically significant. n.s.—no significance. n.d.—not determined.

**Table 10 ijms-25-05995-t010:** Genotypic frequencies of selected SNPs of the *BRCA1* gene in different populations and their associations with the risk of thyroid cancer.

Population	SNP Reference (rs)	Genotype	OR (95% CI)	*p*-Value	References
non-Hispanic white or other	A1988G	AA	1		[78]
		**AG**	**0.63 (0.45–0.87)**	**0.036**	
		GG	1.14 (0.70–1.87)	n.s.	
		**AG + GG**	**0.72 (0.53–0.97)**	**0.026**	
non-Hispanic white or other	rs1799950	**AA**	**1**	**0.017**	[78]
		AG	1.16 (0.72–1.85)	n.s.	
		GG	n.d.	n.d.	
		AG + GG	1.29 (0.82–2.04)	0.34 (n.s.)	
non-Hispanic white or other	rs799917	CC	1	0.22 (n.s.)	[78]
		CT	0.73 (0.52–1.00)	n.s.	
		TT	0.70 (0.45–1.11)	n.s.	
		**CT + TT**	**0.72 (0.53–0.98)**	**0.09**	
non-Hispanic white or other	rs16941	AA	1	0.14 (n.s.)	[78]
		AG	0.71 (0.52–0.98)	n.s.	
		GG	1.11 (0.68–1.82)	n.s.	
		AG + GG	0.78 (0.58–1.05)	0.11 (n.s.)	
Polish	rs16941	AA	1		[80]
		**AG + GG**	**1.31 (1.04–1.67)**	**0.021**	
non-Hispanic white or other	rs16842	AA	1		[78]
		AG	0.68 (0.50–0.94)	n.s.	
		GG	0.94 (0.55–1.60)	n.s.	
		**AG + GG**	**0.72 (0.54–0.98)**	**0.051**	
non-Hispanic white or other	rs1060915	CC	1	0.18 (n.s.)	[78]
		CT	0.74 (0.53–1.03)	n.s.	
		TT	1.11 (0.71–1.75)	n.s.	
		CT + TT	0.82 (0.60–1.11)	0.15 (n.s.)	
non-Hispanic white or other	rs1799966	AA	1	0.32 (n.s.)	[78]
		AG	0.75 (0.54–1.02)	n.s.	
		GG	1.09 (0.65–1.85)	n.s.	
		AG + GG	0.80 (0.59–1.08)	0.17 (n.s.)	
Chinese	rs12516	GG	1		[54]
		AG	0.935 (0.620–1.410)	0.748 (n.s.)	
		AA	1.069 (0.562–2.033)	0.839 (n.s.)	
Chinese	rs8176318	CC	1		[54]
		AC	1.079 (0.715–1.629)	0.717 (n.s.)	
		AA	1.087 (0.570–2.047)	0.800 (n.s.)	

Results in **bold** are statistically significant. n.s.—no significance. n.d.—not determined.

**Table 11 ijms-25-05995-t011:** Genotypic frequencies of a selected SNP of the *BRCA2* gene in the study population and its associations with the risk of thyroid cancer.

Population	SNP Reference (rs)	Genotype	OR (95% CI)	*p*-Value	References
Chinese	rs15869	AA	1		[54]
		AC	1.079 (0.719–1.619)	0.714 (n.s.)	
		**CC**	**2.595 (1.091–6.171)**	**0.031**	

Results in **bold** are statistically significant. n.s.—no significance.

**Table 12 ijms-25-05995-t012:** Genotypic frequencies of selected SNPs of the *APE1* gene in different populations and their associations with the risk of thyroid cancer.

Population	SNP Reference (rs)	Genotype	OR (95% CI)	*p*-Value	References
Chinese	rs3136820	AA	1		[30]
		AG	1.08 (0.78–1.49)	n.s.	
		GG	1.17 (0.76–1.80)	n.s.	
Portuguese	rs3136820	AA	1		[35]
		AG	0.91 (0.52–1.60)	n.s.	
		GG	0.80 (0.43–1.52)	n.s.	
Kazakh and Russian	rs1130409	DD	1		[83]
		DE	1.0 (0.4–2.6)	n.s.	
		EE	1.4 (0.5–4.1)	n.s.	

n.s.—no significance.

**Table 13 ijms-25-05995-t013:** Genotypic frequencies of selected SNPs of the *PARP1* gene in different populations and their associations with the risk of thyroid cancer.

Population	SNP Reference (rs)	Genotype	OR (95% CI)	*p*-Value	References
Chinese	rs1136410	AA	1		[30]
		AG	1.21 (0.87–1.70)	n.s.	
		GG	1.39 (0.91–2.14)	n.s.	
Portuguese	rs1136410	AA	1		[35]
		AG	0.79 (0.46–1.35)	n.s.	
Pakistani	rs1136410	TT	1		[87]
		TC	1.17 (0.85–1.62)	0.31 (n.s.)	
		**CC**	**1.30 (0.99–1.71)**	**0.05**	
		CC + TC	1.36 (0.97–1.89)	0.06 (n.s.)	
Pakistani	rs1805414	TT	1		[87]
		**TC**	**0.80 (0.65–1.00)**	**0.05**	
		CC	1.29 (0.94–1.77)	0.1 (n.s.)	
		**CC + TC**	**0.43 (0.27–0.67)**	**0.003**	
Pakistani	rs1805404	CC	1		[87]
		CT	0.93 (0.69–1.27)	0.69 (n.s.)	
		**TT**	**0.63 (0.40–1.00)**	**0.05**	
		TT + CT	0.77 (0.57–1.02)	0.07 (n.s.)	

Results in **bold** are statistically significant. n.s.—no significance.

**Table 14 ijms-25-05995-t014:** Genotypic frequencies of selected SNPs of the *OGG1* gene in different populations and their associations with the risk of thyroid cancer.

Population	SNP Reference (rs)	Genotype	OR (95% CI)	*p*-Value	References
Spanish	rs1052133	CC	1		[36]
		CG	0.97 (0.72–1.31)	n.s.	
		GG	1.28 (0.65–2.51)	n.s.	
Portuguese	rs1052133	CC	1		[35]
		CG	0.72 (0.43–1.19)	n.s.	

n.s.—no significance.

**Table 15 ijms-25-05995-t015:** Genotypic frequencies of a selected SNP of the *MUTYH* gene in the study population and its associations with the risk of thyroid cancer.

Population	SNP Reference (rs)	Genotype	OR (95% CI)	*p*-Value	References
Portuguese	rs3219489	CC	1		[35]
		CG	0.68 (0.42–1.10)	n.s.	
		GG	1.14 (0.35–3.73)	n.s.	

n.s.—no significance.

**Table 16 ijms-25-05995-t016:** Genotypic frequencies of selected SNPs of the *ATM* gene in different populations and their associations with the risk of thyroid cancer.

Population	SNP Reference (rs)	Genotype	OR (95% CI)	*p*-Value	References
Polish	rs1801516	GG	1		[80]
		GA + AA	0.82 (0.46–1.46)	0.499 (n.s.)	
French Polynesian	rs1801516	GG	1		[98]
		**GA**	**3.13 (1.17**–**8.31)**	**0.02**	
Cuban	rs1801516	GG	1		[99]
		GA	1.2 (0.7–2.0)	0.8 (n.s.)	
		AA	n.d.	n.d.	
non-Hispanic white	rs1801516	GG	1		[97]
		GA + AA	0.9 (0.7–1.1)	n.s.	
Russian and Belarus	rs1801516	GG	1		[71]
		GA	0.75 (0.49–1.15)	0.31 (n.s.)	
		AA	0.61 (0.21–1.77)	0.45 (n.s.)	
Chinese	rs664677	CC + TT	1		[95]
		CT	1.15 (0.86–1.55)	0.34 (n.s.)	
Russian and Belarus	rs664677	TT	1		[71]
		TC	1.03 (0.70–1.50)	0.74 (n.s.)	
		CC	1.19 (0.70–2.04)	0.47 (n.s.)	
Chinese	rs373759	GG + AA	1		[95]
		**AG**	**1.38 (1.03–1.87)**	**0.03**	
Chinese	rs4988099	AA + GA	1		[95]
		GG	0.49 (0.04–5.47)	0.55 (n.s.)	
Chinese	rs189037	GG	1		[95]
		GA + AA	1.17 (0.84–1.64)	0.34 (n.s.)	
non-Hispanic white	rs189037	AA	1		[97]
		**AG + GG**	**0.8 (0.6–1.0)**	**0.04**	
Russian and Belarus	rs609429	CC	1		[71]
		CG	1.10 (0.75–1.62)	0.55 (n.s.)	
		GG	1.14 (0.69–1.89)	0.84 (n.s.)	
non-Hispanic white	rs228589	TT	1		[97]
		TA + AA	0.8 (0.6–1.0)	n.s.	
non-Hispanic white	rs1800054	CC	1		[97]
		CG	1.6 (0.8–3.4)	n.s.	
non-Hispanic white	rs4986761	TT	1		[97]
		TC	1.2 (0.6–2.4)	n.s.	
non-Hispanic white	rs1800057	CC	1		[97]
		**CG + GG**	**1.9 (1.1–3.1)**	**0.02**	

Results in **bold** are statistically significant. n.s.—no significance. n.d.—not determined.

**Table 17 ijms-25-05995-t017:** Genotypic frequencies of a selected SNP of the *CHK2* gene in the study population and its associations with the risk of thyroid cancer.

Population	SNP Reference (rs)	Genotype	OR (95% CI)	*p*-Value	References
Polish	rs17879961	CC	1		[80]
		**CT + CC**	**2.21 (1.69–2.88)**	**9.60 × 10^−10^**	

Results in **bold** are statistically significant.

## Data Availability

Not applicable.

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
