# Peer review of "Polymorphisms of DNA Repair Genes in Thyroid Cancer"

_ijms, 2024, doi:10.3390/ijms25115995_

Round 1

Reviewer 1 Report

Comments and Suggestions for Authors

In this review the authors analyzed the polymorphisms of DNA damage repair genes in thyroid cancer. The article is well written, collects relevant information and analyzes it, which is valuable as well, what is interesting is that they make a comparison between different populations, Pakistani, Chinese, Spanish, Non-Hispanic white, etc.

Collecting and analyzing this information is a great job, but they do not mention if there is no information from other populations and that is why they did not include or why they only analyze the populations they list, it would be valuable to mention it.

Also, figure 1 is very well done but it is pixelated, it would be good to improve the quality.

Reviewer 2 Report

Comments and Suggestions for Authors

The manuscript entitled "Polymorphisms of DNA repair genes in thyroid cancer" is a well-written and -organized review paper, which summarizes all the literature regarding the polymorphisms of the most common DDR- and DNA repair-associated molecules  in thyroid cancer.

The manuscript is worth publishing iN IJMS, however I have one suggestion that may increase its overall presentation. 

I think Figure 1 is not so clear for readers. You could change its orientation  or you could create a new more clear.

Reviewer 3 Report

Comments and Suggestions for Authors

The increasing incidence of thyroid cancer globally highlights the need to understand its risk factors, including the role of DNA repair gene polymorphisms. These polymorphisms can impact genome stability and increase susceptibility to thyroid cancer. Significant differences in the frequency of these polymorphisms across various populations suggest potential disparities in disease risk. Understanding these genetic variations can inform prevention strategies and the development of targeted therapies. The article reviews common polymorphisms in DNA repair genes linked to thyroid cancer and emphasizes the importance of further research. Future studies should explore less investigated DNA repair genes and conduct large-scale, multi-ethnic analyses. This research could fill knowledge gaps and contribute to more effective thyroid cancer management.

Major Comments:

1. Authors should add an additional paragraph for low-fidelity polymerases. Their regulation with different DNA repair genes.

2. Authors should discuss how low-fidelity polymerases play vital role in therapeutic resistance.

3. Authors should use examples of lung and colon cancer in regulating drug resistance

4. Authors should discuss the role of receptors (EGFR / AXL) in regulating DNA repiar genes, high and low fidelity polymerases.

5. Authors should introduce the external agents that could lead to polymorphism that alters the regulation of DNA repair genes.

6. Authors should discuss the compensation between low and high-fidelity polymerases in regulating the resistance of cancer cells to targeted therapy.

Round 2

Reviewer 3 Report

Comments and Suggestions for Authors

The authors have addressed my comments. However, they should cite important articles in contributing to the field. 

Minor Comments: 

Please cite them in the manuscript. 

Authors have to include missing references. 

1.  PMCID: PMC7428051

2. PMCID: PMC4693249

3. PMCID: PMC9627128

4. PMCID: PMC7748119

5. PMCID: PMC8266832

6. PMCID: PMC7303094

7. PMCID: PMC7721097
